# Monotonicity and Double Descent in Uncertainty Estimation with Gaussian Processes

## Abstract

The quality of many modern machine learning models improves as model complexity increases, an effect that has been quantified—for predictive performance—with the non-monotonic double descent learning curve. Here, we address the overarching question: is there an analogous theory of double descent for models which estimate uncertainty? We provide a partially affirmative and partially negative answer in the setting of Gaussian processes (GP). Under standard assumptions, we prove that higher model quality for optimally-tuned GPs (including uncertainty prediction) under marginal likelihood is realized for larger input dimensions, and therefore exhibits a monotone error curve. After showing that marginal likelihood does not naturally exhibit double descent in the input dimension, we highlight related forms of posterior predictive loss that do exhibit non-monotonicity. Finally, we verify empirically that our results hold for real data, beyond our considered assumptions, and we explore consequences involving synthetic covariates.

## 1 Introduction

With the recent success of overparameterized and nonparametric models for many predictive tasks in machine learning (ML), the development of the corresponding uncertainty quantification (UQ) has unsurprisingly become a topic of significant interest. Naïve approaches for forward propagation of error and other methods for inverse uncertainty problems typically apply Monte Carlo methods under a Bayesian framework (Zhang, 2021). However, the large-scale nature of many ML problems of interest results in significant computational challenges. One of the most successful approaches for solving inverse uncertainty problems is the use of *Gaussian processes* (GP) (Williams & Rasmussen, 2006). This is now frequently used for many predictive tasks, including time-series analysis (Roberts et al., 2013) and classification (Williams & Barber, 1998; Williams & Rasmussen, 2006). GPs are also fundamental for Bayesian optimization (Hebbal et al., 2019; Frazier, 2018), although extending Bayesian optimization into high-dimensional settings remains challenging (Binois & Wycoff, 2021).

Although the theoretical understanding of the predictive capacity of high-dimensional ML models continues to advance rapidly, a parallel rigorous theory for UQ is comparatively lagging. The prominent heuristic in modern ML that larger models will typically perform better has become almost axiomatic. However, it is only more recently that this heuristic has become represented in the theory through the characterisation of benign overfitting (Bartlett et al., 2020). In particular, the *double descent* curve extends the bias-variance tradeoff curve to account for improving performance with higher model complexity (Belkin et al., 2019; Wang et al., 2021; Derezinski et al., 2020b) (see Figure 1(right)). Typically, these arguments involve applications of random matrix theory (Edelman & Rao, 2005; Paul & Aue, 2014), notably the Marchenko-Pastur law, concerning limits of spectral distributions under large data/large dimension regimes.

While the predictive performance of ML models can improve as model size increases, it is not clear whether or not the same is true for predictions of model uncertainty. Several common measures of model quality which incorporate inverse uncertainty quantification are Bayesian in nature, the most prominent of which are the *marginal likelihood* and various forms of posterior predictive loss. It is well-known that Bayesian methods can perform well in high dimensions (De Roos et al., 2021), even outperforming their low-dimensional counterparts when properly tuned (Wilson & Izmailov, 2020). To close this theory-practice gap, an analogous formulation of double descent curves in the setting of uncertainty quantification is desired. Marginal likelihood and posterior distributions are often

| Performance Metric | Error Curve | Optimal $\gamma$ |
|---|---|---|
| Marginal Likelihood / Free Energy (3) | Monotone (Thm. 1) | eqn. (5) |
| Posterior Predictive $L^2$ Loss (1) | Double Descent (Prop. 1) | 0 |
| Posterior Predictive NLL (2) | Double Descent (Prop. 1) | $\mathbb{E}\|\bar{f}(x) - y\|^2$ |

Table 1: Behavior of UQ performance metrics and optimal posterior temperature $\gamma$.

intractable for arbitrary models (e.g., Bayesian neural networks (Goan & Fookes, 2020)). However, their explicit forms are well known for GPs (Williams & Rasmussen, 2006).

GPs are nonparametric, and most of the kernels used in practice induce infinite-dimensional feature spaces, so model complexity can be difficult to quantify (although some notions of kernel dimension have been proposed (Zhang, 2005; Alaoui & Mahoney, 2015)). Nevertheless, it is generally expected that accurately fitting a GP to data lying in higher-dimensional spaces requires training on a larger dataset. This *curse of dimensionality* has been justified using error estimates (von Luxburg & Bousquet, 2004), and verified empirically (Spigler et al., 2020). However, under appropriate setups, predictive performance has been demonstrated to *improve* with larger input dimension (Liu et al., 2021). Here, we consider whether the same is true for the marginal likelihood and posterior predictive metrics. Our main results (see Theorem 1 and Proposition 1) are summarized as follows.

- ***Monotonicity***: *For two optimally regularized scalar GPs, both fit to a sufficiently large set of iid normalized and whitened input-output pairs, the better performing model under marginal likelihood is the one with larger input dimension.*

- ***Double Descent***: *For sufficiently small temperatures, GP posterior predictive metrics exhibit double descent if and only if the mean squared error for the corresponding kernel regression task exhibits double descent (see Liang & Rakhlin (2020); Liu et al. (2021) for sufficient conditions).*

Figure 1 illustrates characteristics of monotone and double descent error curves. Along the way, we identify optimal choices of temperature (which can be interpreted as noise in the data) under a tempered posterior setup — see Table 1 for a summary. Our results highlight that the common curse of dimensionality heuristic can be bypassed through an empirical Bayes procedure. Furthermore, the performance of optimally regularized GPs (under several metrics), can be improved with additional covariates (including synthetic ones). Our theory is supported by experiments performed on real large datasets. Additional experiments, including the effect of ill-conditioned inputs, alternative data distributions, and choice of underlying kernel, are conducted in Appendix A. Details of the setup for each experiment are listed in Appendix G.

## 2 BACKGROUND

### 2.1 GAUSSIAN PROCESSES

A *Gaussian process* is a stochastic process $f$ on $\mathbb{R}^d$ such that for any set of points $x_1, \ldots, x_k \in \mathbb{R}^d$, $(f(x_1), \ldots, f(x_k))$ is distributed as a multivariate Gaussian random vector (Williams & Rasmussen, 2006, §2.2). Gaussian processes are completely determined by their *mean* and *covariance functions*: if for any $x, x' \in \mathbb{R}^d$, $\mathbb{E}f(x) = m(x)$ and $\text{Cov}(f(x), f(x')) = k(x, x')$, then we say that $f \sim \mathcal{GP}(m, k)$. Inference for GPs is informed by Bayes' rule: letting $(X_i, Y_i)_{i=1}^n$ denote a collection of iid input-output pairs, we impose the assumption that $Y_i = f(X_i) + \epsilon_i$ where each $\epsilon_i \sim \mathcal{N}(0, \gamma)$,

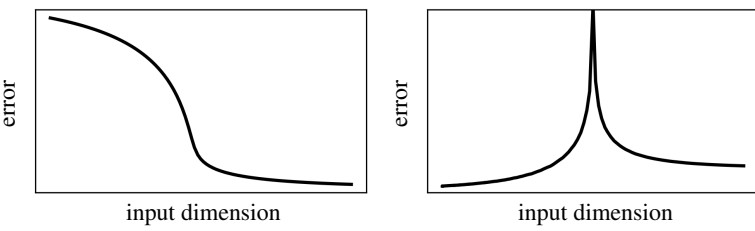

Figure 1: Illustrations of monotone (left) and double descent (right) error curves.

yielding a Gaussian likelihood $p(Y|f, X) = (2\pi\gamma)^{-n/2} \exp(-\frac{1}{2\gamma}\|Y - f(X)\|^2)$. The parameter $\gamma$ is the *temperature* of the model, and dictates the perceived accuracy of the labels. For example, taking $\gamma \to 0^+$ considers a model where the labels are noise-free.

For the prior, we assume that $f \sim \mathcal{GP}(0, \lambda^{-1}k)$ for a fixed covariance kernel $k$ and regularization parameter $\lambda > 0$. While other mean functions $m$ can be considered, in the sequel we will consider the case where $m \equiv 0$. Indeed, if $m \neq 0$, then one can instead consider $\tilde{Y}_i = Y_i - m(X_i)$, so that $\tilde{Y}_i = \tilde{f}(X_i) + \epsilon_i$ and the corresponding prior for $\tilde{f}$ is zero-mean. The Gram matrix $K_X \in \mathbb{R}^{n \times n}$ for $X$ has elements $K_X^{ij} = k(X_i, X_j)$. Let $\boldsymbol{x} = (x_1, \ldots, x_m)$ denote a collection of $N$ points in $\mathbb{R}^d$, $f(\boldsymbol{x}) = (f(x_1), \ldots, f(x_m))$ and denote by $K_{\boldsymbol{x}} \in \mathbb{R}^{m \times m}$ and $k_{\boldsymbol{x}} \in \mathbb{R}^{n \times m}$ the matrices with elements $K_{\boldsymbol{x}}^{ij} = k(x_i, x_j)$ and $k_{\boldsymbol{x}}^{ij} = k(X_i, x_j)$.

Given this setup, we are interested in several metrics which quantify the uncertainty of the model. The **posterior predictive distribution** (PPD) of $f(\boldsymbol{x})$ given $X, Y$ is (Williams & Rasmussen, 2006, pg. 16)

$$f(\boldsymbol{x})|X, Y \sim \mathcal{N}(\bar{f}(\boldsymbol{x}), \lambda^{-1}\Sigma(\boldsymbol{x})),$$

where $\bar{f}(\boldsymbol{x}) = k_{\boldsymbol{x}}^\top (K_X + \lambda\gamma I)^{-1}Y$ and $\Sigma(\boldsymbol{x}) = K_{\boldsymbol{x}} - k_{\boldsymbol{x}}^\top(K_X + \lambda\gamma I)^{-1}k_{\boldsymbol{x}}$. This defines a posterior predictive distribution $\rho^\gamma$ on the GP $f$ given $X, Y$ (so $f \mid X, Y \sim \rho^\gamma$). If we let $\boldsymbol{y} = (y_1, \ldots, y_m)$ denote a collection of $m$ associated *test labels* corresponding to our test data $\boldsymbol{x}$, the **posterior predictive $L^2$ loss** (PPL2) is the quantity

$$\ell(\boldsymbol{x}, \boldsymbol{y}) := \mathbb{E}_{f \sim \rho^\gamma}\|f(\boldsymbol{x}) - \boldsymbol{y}\|^2 = \|\bar{f}(\boldsymbol{x}) - \boldsymbol{y}\|^2 + \lambda^{-1}\mathrm{tr}(\Sigma(\boldsymbol{x})). \tag{1}$$

Closely related is the **posterior predictive negative log-likelihood** (PPNLL), given by

$$L(\boldsymbol{x}, \boldsymbol{y}|X, Y) := -\mathbb{E}_{f \sim \rho^\gamma} \log p(\boldsymbol{y}|f, \boldsymbol{x}) = \frac{1}{2\gamma}\|\bar{f}(\boldsymbol{x}) - \boldsymbol{y}\|^2 + \frac{1}{2\lambda\gamma}\mathrm{tr}(\Sigma(\boldsymbol{x})) + \frac{m}{2}\log(2\pi\gamma). \tag{2}$$

## 2.2 Marginal Likelihood

The fundamental measure of model performance in Bayesian statistics is the *marginal likelihood* (also known as the *partition function* in statistical mechanics). Integrating the likelihood over the prior distribution $\pi$ provides a probability density of data under the prescribed model. Evaluating this density at the training data gives an indication of model suitability before posterior inference. Hence, the marginal likelihood is $\mathcal{Z}_n = \mathbb{E}_{f \sim \pi}p(Y|f, X)$. A larger marginal likelihood is typically understood as an indication of superior model quality. The **Bayes free energy** $\mathcal{F}_n = -\log \mathcal{Z}_n$ is interpreted as an analogue of the test error, where smaller $\mathcal{F}_n$ is desired.

The **marginal likelihood for a Gaussian process** is straightforward to compute: since $Y_i = f(X_i) + \epsilon_i$ under the likelihood, and $(f(X_i))_{i=1}^n \sim \mathcal{N}(0, \lambda^{-1}K_X)$ under the GP prior $\pi = \mathcal{GP}(0, \lambda^{-1}k)$, we have $Y_i|X \sim \mathcal{N}(0, \lambda^{-1}K_X + \gamma I)$, and hence the Bayes free energy is given by (Williams & Rasmussen, 2006, eqn. (2.30))

$$\mathcal{F}_n^\gamma = \frac{1}{2}\lambda Y^\top(K_X + \lambda\gamma I)^{-1}Y + \frac{1}{2}\log\det(K_X + \lambda\gamma I) - \frac{n}{2}\log\left(\frac{\lambda}{2\pi}\right). \tag{3}$$

In practice, the hyperparameters $\lambda, \gamma$ are often tuned to minimize the Bayes free energy. This is an *empirical Bayes procedure*, and typically achieves excellent results (Krivoruchko & Gribov, 2019).

The PPNLL can be linked to the marginal likelihood through cross-validation measures. Let $I$ be uniform on $\{1, \ldots, k\}$ and let $\mathcal{T}$ be a random choice of $k$ indices from $\{1, \ldots, n\}$ (the *test set*). Let $\bar{\mathcal{T}} = \{1, \ldots, n\}\backslash\mathcal{T}$ denote the corresponding *training set*. The leave-$k$-out cross-validation score under the PPNLL is defined by $S_k(X, Y) = \mathbb{E}L(X_{\mathcal{T}_I}, Y_{\mathcal{T}_I}|X_{\bar{\mathcal{T}}}, Y_{\bar{\mathcal{T}}})$. The Bayes free energy is the sum of all leave-$k$-out cross-validation scores (Fong & Holmes, 2020), that is $\mathcal{F}_n^\gamma = \sum_{k=1}^n S_k(X, Y)$. Therefore, the **mean Bayes free energy** (or mean free energy for brevity) $n^{-1}\mathcal{F}_n^\gamma$ can be interpreted as the average cross-validation score. Similar connections can also be formulated in the PAC-Bayes framework (Germain et al., 2016).

## 2.3 Bayesian Linear Regression

One of the most important special cases of GP regression is *Bayesian linear regression*, obtained by taking $k_{\mathrm{lin}}(x, x') = x^\top x'$. As a special case of GPs, our results apply to Bayesian linear regression, directly extending double descent analysis into the Bayesian setting. By Mercer's Theorem

(Williams & Rasmussen, 2006, §4.3.1), a realization of a GP $f$ has a series expansion in terms of the eigenfunctions of the kernel $k$. As the eigenfunctions of $k_{\text{lin}}$ are linear, $f \sim \mathcal{GP}(0, \lambda^{-1}k_{\text{lin}})$ if and only if

$$f(x) = w^\top x, \qquad w \sim \mathcal{N}(0, \lambda^{-1}).$$

More generally, if $\phi : \mathbb{R}^d \to \mathbb{R}^p$ is a finite-dimensional feature map, then $f(x) = w^\top \phi(x)$, $w \sim \mathcal{N}(0, \lambda^{-1})$ is a GP with covariance kernel $k_\phi(x, y) = \phi(x)^\top \phi(y)$. This is the weight-space interpretation of Gaussian processes. In this setting, the posterior distribution over the weights satisfies $\rho^\gamma(w) = p(w|X, Y) \propto \exp(-\frac{1}{2\gamma}\|Y - \phi(X)w\|^2 - \frac{\lambda}{2}\|w\|^2)$ and the marginal likelihood becomes

$$\mathcal{Z}_n^\gamma = \int_{\mathbb{R}^p} p(Y|X, w)\pi(w)\mathrm{d}w = \frac{\lambda^{d/2}}{\gamma^{n/2}(2\pi)^{\frac{1}{2}(n+d)}} \int_{\mathbb{R}^p} e^{-\frac{1}{2\gamma}\|Y - \phi(X)w\|^2} e^{-\frac{\lambda}{2}\|w\|^2}\mathrm{d}w, \qquad (4)$$

where $\phi(X) = (\phi(X_i))_{i=1}^n \in \mathbb{R}^{n \times p}$. Under this interpretation, the role of $\lambda$ as a regularization parameter is clear. Note also that if $\lambda = \mu/\gamma$ for some $\mu > 0$, then the posterior $\rho^\gamma(w)$ depends on $\gamma$ as $(\rho^1(w))^{1/\gamma}$ (excluding normalizing constants). This is called a *tempered posterior*; if $\gamma < 1$, the posterior is *cold*, and it is *warm* whenever $\gamma > 1$.

## 3 RELATED WORK

**Double Descent and Multiple Descent.**   *Double descent* is an observed phenomenon in the error curves of kernel regression, where the classical (U-shaped) bias-variance tradeoff in underparameterized regimes is accompanied by a curious monotone improvement in test error as the ratio $c$ of the number of datapoints to the ambient data dimension increases beyond $c = 1$. The term was popularized in Belkin et al. (2018b; 2019). However, it had been observed in earlier reports (Dobriban & Wager, 2018; Loog et al., 2020), and the existence of such non-monotonic behavior as a function of system control parameters should not be unexpected, given general considerations about different phases of learning that are well-known from the statistical mechanics of learning (Engel & den Broeck, 2001; Martin & Mahoney, 2017). An early precursor to double descent analysis came in the form of the *Stein effect*, which establishes uniformly reduced risk when some degree of regularisation is added (Strawderman, 2021). Stein effects have been established for kernel regression in Muandet et al. (2014); Chang et al. (2017). Subsequent theoretical developments proved the existence of double descent error curves on various forms of linear regression (Bartlett et al., 2020; Tsigler & Bartlett, 2020; Hastie et al., 2022; Muthukumar et al., 2020), random features models (Liao et al., 2020; Holzmüller, 2020), kernel regression (Liang & Rakhlin, 2020; Liu et al., 2021), two-layer neural networks (Mei & Montanari, 2022), and classification tasks (Frei et al., 2022; Wang et al., 2021). For non-asymptotic results, subgaussian data is commonly assumed, yet other data distributions have also been considered (Derezinski et al., 2020b). Double descent error curves have also been observed in nearest neighbor models (Belkin et al., 2018a), decision trees (Belkin et al., 2019), and state-of-the-art neural networks (Nakkiran et al., 2021). More recent developments have identified a large number of possible curves in kernel regression (Liu et al., 2021), including triple descent (Adlam & Pennington, 2020; d'Ascoli et al., 2020) and multiple descent for related volume-based metrics (Derezinski et al., 2020a). Similar to our results, an optimal choice of regularization parameter can negate the double descent singularity and result in a monotone error curve (Liu et al., 2021; Nakkiran et al., 2020; Wu & Xu, 2020). While there does not appear to be clear consensus on a *precise* definition of "double descent," for our purposes, we say that an error curve $E(t)$ exhibits double descent if it contains a single global maximum away from zero at $t^*$, and decreases monotonically thereafter. This encompasses double descent as it appears in the works above, while excluding some misspecification settings and forms of multiple descent.

**Learning Curves for Gaussian Processes.**   The study of error curves for GPs under posterior predictive losses has a long history (see Williams & Rasmussen (2006, §7.3) and Viering & Loog (2021)). However, most results focus on rates of convergence of posterior predictive loss in the large data regime $n \to \infty$. The resulting error curve is called a *learning curve*, as it tracks how fast the model learns with more data (Sollich, 1998; Sollich & Halees, 2002; Le Gratiet & Garnier, 2015). Of particular note are classical upper and lower bounds on posterior predictive loss (Opper & Vivarelli, 1998; Sollich & Halees, 2002; Williams & Vivarelli, 2000), which are similar in form to counterparts in the double descent literature (Holzmüller, 2020). For example, some upper bounds have been obtained with respect to forms of *effective dimension*, defined in terms of the Gram matrix

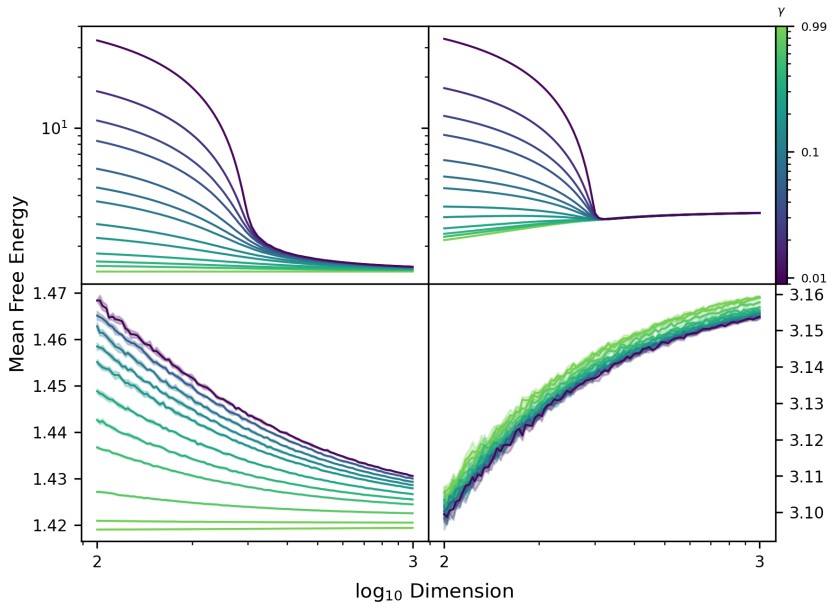

Figure 2: Error curves for mean Bayes free energy $n^{-1}\mathcal{F}_n^\gamma$ for **synthetic data** under linear (top) and Gaussian (bottom) kernels, with $\lambda = \lambda^*$ (left; **monotone decreasing**) and $\lambda = 0.01$ (right; **increases at higher input dimensions**).

(Zhang, 2005; Alaoui & Mahoney, 2015). Contraction rates in the posterior have also been examined (Lederer et al., 2019). In our work, we consider error curves over dimension rather than data, but we note that our techniques could also be used to study learning curves.

**Cold Posteriors.** Among the most surprising phenomena encountered in Bayesian deep learning is the *cold posterior effect* (CPE): the performance of Bayesian neural networks (BNNs) appears to improve for tempered posteriors when $\gamma \to 0^+$. This presents a challenge for uncertainty prediction: taking $\gamma \to 0^+$ concentrates the posterior around the *maximum a posteriori* (MAP) point estimator, and so the CPE implies that optimal performance is achieved when there is little or no predicted uncertainty. First observed in Wenzel et al. (2020), several authors have since sought to explain the phenomenon through data curation (Aitchison, 2020), data augmentation (Izmailov et al., 2021; Fortuin et al., 2021), and misspecified priors (Wenzel et al., 2020), although the CPE can still arise in isolation of each of these factors (Noci et al., 2021). While our setup is too simple to examine the CPE at large, we find some common forms of posterior predictive loss are optimized as $\gamma \to 0^+$.

## 4 MONOTONICITY IN BAYES FREE ENERGY

In this section, we investigate the behavior of the Bayes free energy using the explicit expression in (3). First, to facilitate our analysis, we require the following assumption on the kernel $k$.

**Assumption.** *The kernel $k$ is formed by a function $\kappa : \mathbb{R} \to \mathbb{R}$ that is continuously differentiable on $(0, \infty)$ and is one of the following two types:*

(I) ***Inner product kernel:*** *$k(x, x') = \kappa(x^\top x'/d)$ for $x, x' \in \mathbb{R}^d$, where $\kappa$ is three-times continuously differentiable in a neighbourhood of zero, with $\kappa'(0) > 0$. Let*

$$\alpha = \kappa'(0), \qquad \beta = \kappa(1) - \kappa(0) - \kappa'(0).$$

(II) ***Radial basis kernel:*** *$k(x, x') = \kappa(\|x - x'\|^2/d)$ for $x, x' \in \mathbb{R}^d$, where $\kappa$ is three-times continuously differentiable on $(0, \infty)$, with $\kappa'(2) < 0$. Let*

$$\alpha = -2\kappa'(2), \qquad \beta = \kappa(0) + 2\kappa'(2) - \kappa(2).$$

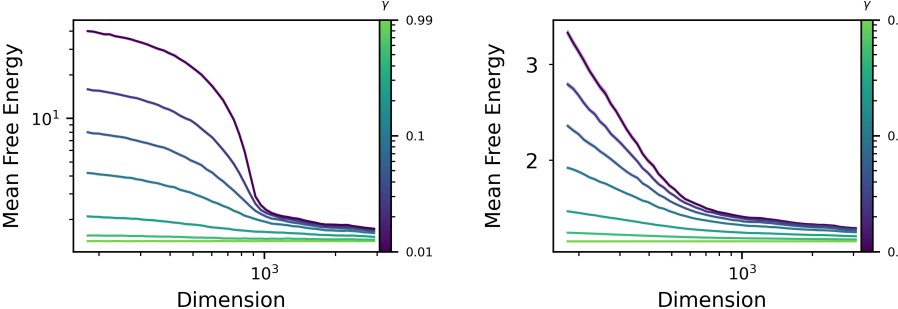

Figure 3: Error curves for mean Bayes free energy under the CIFAR10 dataset; linear (left) and Gaussian (right) kernels; $\lambda = \lambda^*$; **curves for real data match Figure 2 (left)**.

This assumption allows for many common covariance kernels used for GPs, including polynomial kernels $k(x, x') = (c + x^\top x'/d)^p$, the exponential kernel $k(x, x') = \exp(x^\top x'/d)$, the Gaussian kernel $k(x, x') = \exp(-\|x - x'\|^2/d)$, the multiquadric kernel $k(x, x') = (c + \|x - x'\|^2/d)^p$, the inverse multiquadric $k(x, x') = (c + \|x - x'\|^2/d)^{-p}$ kernels, and the Matérn kernels

$$k(x, x') \;=\; (2^{\nu-1}\Gamma(\nu))^{-1}\|x \;-\; x'\|^\nu K_\nu(\|x - x'\|)$$

(where $K_\nu$ is the Bessel-$K$ function). Different bandwidths can also be incorporated through the choice of $\kappa$. However, it does exclude some of the more recent and sophisticated kernel families, e.g., neural tangent kernels. Due to a result of El Karoui (2010), the Gram matrices of kernels satisfying this assumption exhibit limiting spectral behavior reminiscent of that for the linear kernel, $k(x, x') = c + x^\top x'/d$. Roughly speaking, from the perspective of the marginal likelihood, we can treat GPs as Bayesian linear regression.

In line with previous work on double descent curves (Belkin et al., 2019), our objective is to investigate the behavior of the marginal likelihood with respect to model complexity, which is often given by the number of parameters in parametric settings (d'Ascoli et al., 2020; Derezinski et al., 2020b; Hastie et al., 2022)). GPs are non-parametric, and while notions of *effective dimension* do exist (Zhang, 2005; Alaoui & Mahoney, 2015), it is common to instead consider the input dimension in place of the number of parameters in this context (Liang & Rakhlin, 2020; Liu et al., 2021).

For our theory, we first consider the "best-case scenario," where the prior is perfectly specified and its mean function $m$ is used to generate $Y$: $Y_i = m(X_i) + \epsilon_i$, where each $\epsilon_i$ is iid with zero mean and unit variance. By a change of variables, we can assume (without loss of generality) that $m \equiv 0$, so that $Y_i = \epsilon_i$, and is therefore independent of $X$. To apply the Marchenko-Pastur law from random matrix theory, we consider the large dataset – large input dimension limit, where $n$ and $d$ scale linearly so that $d/n \to c \in (0, \infty)$. The inputs are assumed to have been *whitened* and to be independent zero-mean random vectors with unit covariance. Under this limit, the sequence of mean Bayes entropies $n^{-1}\mathcal{F}_n^\gamma$, for each $n = 1, 2, \ldots$, converges in expectation over the training set to a quantity $\mathcal{F}_\infty^\gamma$ which is more convenient to study. Our main result is presented in Theorem 1; the proof is delayed to Supplementary Material.

**Theorem 1** (Limiting Bayes Free Energy). *Let $X_1, X_2, \ldots$ be independent and identically distributed zero-mean random vectors in $\mathbb{R}^d$ with unit covariance, satisfying $\mathbb{E}\|X_i\|^{5+\delta} < +\infty$ for some $\delta > 0$. For each $n = 1, 2, \ldots$, let $\mathcal{F}_n^\gamma$ denote (3) applied to $X = (X_i)_{i=1}^n$ and $Y = (Y_i)_{i=1}^n$, with each $Y_i \sim \mathcal{N}(0, 1)$. If $n, d \to \infty$ such that $d/n \to c \in (0, \infty)$, then*

$$\mathcal{F}_\infty^\gamma \;:=\; \lim_{n\to\infty} n^{-1}\mathbb{E}\mathcal{F}_n^\gamma,$$

*is well-defined. In this case,*

*(a) If $\lambda = \mu/\gamma$ for some $\mu > 0$, there exists an optimal temperature $\gamma^*$ which minimizes $\mathcal{F}_\infty^\gamma$, which is given by*

$$\gamma^* = c - 1 - \tfrac{c}{\alpha}(\beta + \mu) + \sqrt{(1 + \tfrac{c}{\alpha}(\beta + \mu + \alpha))^2 - 4c}. \tag{5}$$

*If the kernel $k$ depends on $\lambda$ such that $\alpha$ is constant in $\lambda$ and $\beta = \beta_0\lambda$ for $\beta_0 \in [0, 1)$, then*

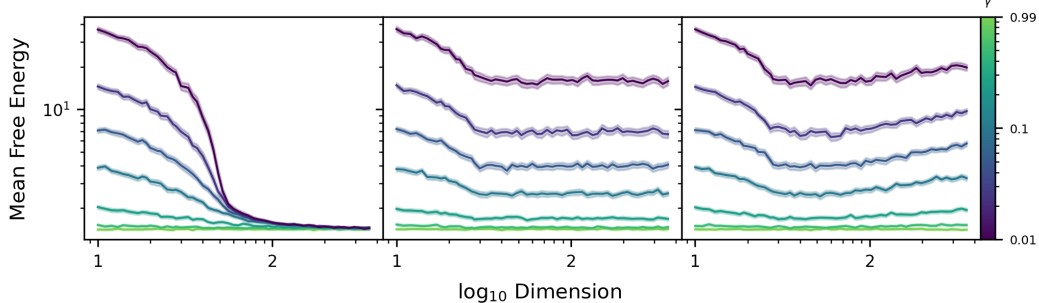

Figure 4: Error curves for mean Bayes free energy under real data with Gaussian (left); repeated data (center); and zeroed data (right), under the linear kernel and $\lambda = \lambda^*$. **Only adding synthetic non-zero iid covariates improves model performance.**

(b) *If* $\gamma \in (0, 1 - \beta_0)$*, there exists a unique optimal* $\lambda^* > 0$ *minimizing* $\mathcal{F}^\gamma_\infty$ *satisfying*

$$\lambda^* = \frac{\alpha[(c+1)(\gamma + \beta_0) + \sqrt{(c-1)^2 + 4c(\gamma + \beta_0)^2}]}{c(1 - (\gamma + \beta_0)^2)}. \qquad (6)$$

*If* $\gamma \geq 1 - \beta_0$*, then no such optimal* $\lambda^*$ *exists.*

(c) *For any temperature* $0 < \gamma < 1 - \beta_0$*, at* $\lambda = \lambda^*$*,* $\mathcal{F}^\gamma_\infty$ *is* **monotone decreasing** *in* $c \in (0, \infty)$*.*

The expression for the asymptotic Bayes free energy $\mathcal{F}^\gamma_\infty$ is provided in the Supplementary Material. To summarize, first, in the spirit of empirical Bayes, there exists an optimal $\lambda^*$ for the Gaussian prior which minimizes the asymptotic mean free energy. Under this setup, the choice of $\lambda$ which maximizes the marginal likelihood for a particular realization of $X, Y$ will converge almost surely to $\lambda^*$ as $n, d \to \infty$. Similar to Nakkiran et al. (2020); Wu & Xu (2020), we find that model performance under marginal likelihood improves monotonically with input dimension when $\lambda = \lambda^*$ for a fixed amount of data. Indeed, for large $n, d$, $\mathbb{E}\mathcal{F}^\gamma_n \approx n\mathcal{F}^\gamma_\infty$ and $c \approx d/n$, so Theorem 1c implies that the expected Bayes free energy decreases (approximately) monotonically with the input dimension, provided $n$ is fixed and the optimal regularizer $\lambda^*$ is chosen.

**Discussion of assumptions.** The assumption that the kernel scales with $\lambda$ is necessary using our techniques, as $\lambda^*$ cannot be computed explicitly otherwise. This holds for the linear kernel, but most other choices of $\kappa$ can be made to satisfy the conditions of Theorem 1 by taking $\kappa(x) \mapsto \eta^{-1}\kappa(\eta x)$, for appropriately chosen bandwidth $\eta \equiv \eta(\lambda)$. For example, for the quadratic kernel, this gives $k(x, x') = (\lambda^{-1/2} + \lambda^{1/2}x^\top x')^2$. Effectively, this causes the regularization parameter to scale non-linearly in the prior kernel. Even though this is required for our theory, we can empirically demonstrate this monotonicity also holds under the typical setup where $k$ does not change with $\lambda$. In Figure 2, we plot the mean free energy for synthetic Gaussian datasets of increasing dimension at both optimal and fixed values of $\lambda$ for the linear and Gaussian kernels. Since $n$ is fixed, in line with Theorem 1c, the curves with optimally chosen $\lambda$ decrease monotonically with input dimension, while the curves for fixed $\lambda$ appear to increase when the dimension is large. Note, however, that the larger $\beta$ for the Gaussian kernel induces a significant regularizing effect. A light CPE appears for the Gaussian kernel when $\lambda$ is fixed, but does not seem to occur under $\lambda^*$.

While the assumption that $m = 0$ may appear too restrictive, in Appendix B, we show that $m$ is necessarily small when the data is normalized and whitened. Consequently, under a zero-mean prior, the marginal likelihood behaves similarly to our assumed scenario. This translates well in practice: under a similar setup to Figure 2, the error curves corresponding to the linear and Gaussian kernels under the whitened `CIFAR10` benchmark dataset (Krizhevsky & Hinton, 2009) exhibiting the predicted monotone behavior (Figure 3).

**Synthetic covariates.** Since Theorem 1 implies that performance under the marginal likelihood can improve as covariates are added, it is natural to ask whether an improvement will be seen if

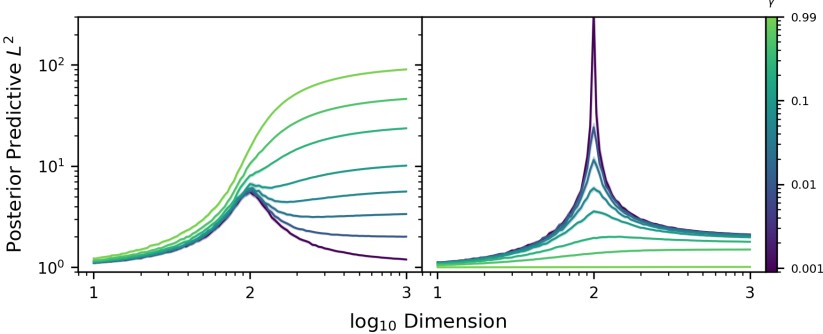

Figure 5: Posterior predictive $L^2$ loss error curves for **synthetic data** exhibiting perturbed / tempered double descent under the linear kernel with $\lambda = 0.01/\gamma$ (left), and $\lambda = \lambda^*$ (right).

the data is augmented with synthetic covariates. To test this, we considered the first 30 covariates of the whitened `CT Slices` dataset obtained from the UCI Machine Learning Repository (Graf et al., 2011), and we augmented them with synthetic (iid standard normal) covariates; the first 30 covariates repeated; and zeros (for more details, see Appendix A). While the first of these scenarios satisfies the conditions of Theorem 1, the second two do not, since the new data cannot be whitened such that its rows have unit covariance. Consequently, the behavior of the mean free energy reflects whether the assumptions of Theorem 1 are satisfied: only the data with Gaussian covariates exhibits the same monotone decay. From a practical point of view, a surprising conclusion is reached: after optimal regularization, performance under marginal likelihood can be further improved by concatenating Gaussian noise to the input.

## 5 DOUBLE DESCENT IN POSTERIOR PREDICTIVE LOSS

In this section, we will demonstrate that, despite the connections between them, the marginal likelihood and posterior predictive loss can exhibit different qualitative behavior, with the posterior predictive losses potentially exhibiting a double descent phenomenon. Observe that the two forms of posterior predictive loss defined in (1) and (2) can both be expressed in the form

$$\mathcal{L} = c_1(\gamma)\underbrace{\mathbb{E}\|\bar{f}(\boldsymbol{x}) - \boldsymbol{y}\|^2}_{\text{MSE}} + c_2(\lambda, \gamma)\underbrace{\mathbb{E}\mathrm{tr}(\Sigma(\boldsymbol{x}))}_{\text{volume}} + c_3(\gamma).$$

The first term is the mean-squared error (MSE) of the predictor $\bar{f}$, and is a well-studied object in the literature. In particular, **the MSE can exhibit double descent**, or other types of multiple descent error curves depending on $k$, in both ridgeless (Holzmüller, 2020; Liang & Rakhlin, 2020) and general (Liu et al., 2021) settings. On the other hand, the volume term has the uniform bound $\mathbb{E}\mathrm{tr}(\Sigma(\boldsymbol{x})) \leq m\mathbb{E}k(x, x)$, so provided $c_2$ is sufficiently small, the volume term should have little qualitative effect. The following is immediate.

**Proposition 1.** *Assume that the MSE $\mathbb{E}\|\bar{f}(\boldsymbol{x}) - \boldsymbol{y}\|^2$ for Gaussian inputs $\boldsymbol{x}$ and labels $\boldsymbol{y}$ converges to an error curve $E(c)$ that exhibits double descent as $n \to \infty$ with $d \equiv d(n)$ satisfying $d(n)/n \to c \in (0, \infty)$. If there exists a function $\lambda(\gamma)$ such that $c_2(\lambda(\gamma), \gamma)/c_1(\gamma) \to 0$ as $\gamma \to 0^+$, then for any $\epsilon > 0$, there exists an error curve $\bar{E}(c)$ exhibiting double descent, a positive integer $N$, and $\gamma_0 > 0$ such that for any $0 < \gamma < \gamma_0$ and $n > N$, $|\mathcal{L}/c_1 - \bar{E}| < \epsilon$ at $d = d(n)$ and $\lambda = \lambda(\gamma)$.*

For **posterior predictive $L^2$ loss**, in the tempered posterior scenario where $\lambda = \mu/\gamma$, the MSE remains constant in $\gamma$, while $c_2/c_1 = \gamma/\mu$. Since the predictor $\bar{f}$ depends only on $\mu$, the optimal $\gamma$ in the tempered posterior scenario is realised as $\gamma \to 0^+$. In other words, under the posterior predictive $L^2$ loss, *the best prediction of uncertainty is none at all*. This highlights a trivial form of CPE for PPL2 losses, suggesting it may not be suitable as a UQ metric. Here we shall empirically examine the linear kernel case; similar experiments for more general kernels are conducted in the Supplementary Material. In Figure 5(left), we plot posterior predictive $L^2$ loss under the linear kernel on synthetic Gaussian data by varying $\gamma$ while keeping $\mu$ fixed. We find that the error curve exhibits double descent when $\gamma < 2\mu$. The corresponding plot for the `CIFAR10` dataset is shown

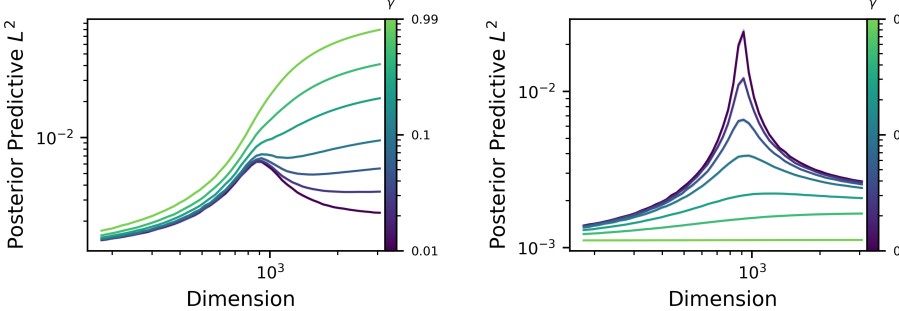

Figure 6: PPL2 loss under the linear kernel with $\lambda = 0.01/\gamma$ (left) and $\lambda = \lambda^*$ (right) on the CIFAR10 dataset; **curves for real data match Figure 5.**

in Figure 6(left), demonstrating that this behavior carries over to real data. Choosing $\lambda = \lambda^*$ (the optimal $\lambda$ according to marginal likelihood) reveals a more typical set of regularized double descent curves; this is shown in Figure 5(right) for synthetic data and Figure 6(right) for the CIFAR10 dataset. This is due to the monotone relationship between the volume term and $\lambda$, hence the error curve inherits its shape from the behaviour of $\lambda^*$ (see Appendix A in the Supplementary Material).

In contrast, this phenomenon is not the case for **posterior predictive negative log-likelihood**. Indeed, letting $\lambda = \mu/\gamma$ and optimizing the expectation of (2) in $\gamma$, the optimal $\gamma^* = m^{-1}\mathbb{E}\|\bar{f}(\boldsymbol{x}) - \boldsymbol{y}\|^2$. The expected optimal PPNLL is therefore

$$-\mathbb{E}_{\boldsymbol{x},\boldsymbol{y}}\mathbb{E}_{f \sim \rho^{\gamma^*}} \log p(\boldsymbol{y}|f, \boldsymbol{x}) = \tfrac{1}{2}m[1 + \log(2\pi\mathbb{E}\|\bar{f}(\boldsymbol{x}) - \boldsymbol{y}\|^2)] + (2\mu)^{-1}\mathrm{tr}(\Sigma(\boldsymbol{x})). \quad (7)$$

Otherwise, the PPNLL displays similar behavior to PPL2, as the two are related linearly.

## 6 CONCLUSION

Motivated by understanding the uncertainty properties of prediction from GP models, we have applied random matrix theory arguments and conducted several experiments to study the error curves of three UQ metrics for GPs. Contrary to classical heuristics, model performance under marginal likelihood/Bayes free energy improves monotonically with input dimension under appropriate regularization (Theorem 1). However, Bayes free energy does not exhibit double descent. Instead, posterior predictive loss inherits a double descent curve from non-UQ settings when the variance in the posterior distribution is sufficiently small (Proposition 1). While our analysis was conducted under the assumption of a perfectly chosen prior mean, similar error curves appear to hold under small perturbations, which always holds for large whitened datasets. Although our contributions are predominantly theoretical, our results also have some noteworthy practical consequences:

- Tuning hyperparameters according to marginal likelihood is **essential** to ensuring good performance in higher dimensions, and **completely negates the curse of dimensionality**.

- When using $L^2$ losses as UQ metrics, care should be taken in view of the CPE. As such, **we do not recommend the use of this metric in lieu of other alternatives**.

- Our experiments suggest that further improvements beyond the optimisation of hyperparameters may be possible with the addition of synthetic covariates, although further investigation is needed before such a procedure can be universally recommended.

In light of the surprisingly complex behavior on display, the fine-scale behavior our results demonstrate, and a surprising absence of UQ metrics in the double descent literature, we encourage increasing adoption of random matrix techniques for studying UQ / Bayesian metrics in double descent contexts and beyond. There are numerous avenues available for future work, including the incorporation of more general kernels (e.g., using results from Fan & Wang (2020) to treat neural tangent kernels, which are commonly used as approximations for large-width neural networks).

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
