# OpenReview forum: "Monotonicity and Double Descent in Uncertainty Estimation with Gaussian Processes"
_ICLR.cc/2023/Conference — Submitted to ICLR 2023_

### Official Review · Reviewer_ZFnV · 2022-10-17

**Confidence:** 4
**Correctness:** 3
**Technical Novelty And Significance:** 2
**Empirical Novelty And Significance:** 2
**Recommendation:** 5

**Clarity, Quality, Novelty And Reproducibility:**

The exposition is overall clear and the mix of formal and heuristic discussion is appropriate.  I do find that the use of the term 'dimension' is confused/confusing in places: primarily the dimension is considered here as the number of spectral features rather than the dimension of the input domain.  The quality is high in terms of technical validity but lower in terms of transferability and applicable insights under the current presentation.  The novelty is difficult to assess as the Stein estimator papers I mentioned have not been discussed with regard to similarities and differences against the present study.  The reproducibility would seem high.


**Strength And Weaknesses:**

The authors demonstrate a strong knowledge of the recent literature in this field.  I have not checked the proofs (though I would happily do so if a revised manuscript is submitted); however, the conditions supposed and the theorem statements themselves are consistent with what I what expect from a proof in their field (i.e., address the natural questions such as differentiability at the origin for the chosen kernels).

However, I struggle to see the value of the theorems given for understanding the real world behaviour of Gaussian process models, owing to the limitations of the problem set up:
- foremost: Unless I have gravely misinterpreted the problem description, the "best-case scenario" (page 6) assumed is that all the data are generated under an iid noise model.  In this case, any improvement that an estimator that correlates the prediction to new locations against data from currently observed locations can give over the naive model (no correlation; modulo that potentially induced by learning a pooled value for the scale of e_i) is the domain of the original Stein effect (Stein 1956).  This has been examined previously for the case of RFF kernel approximations by (e.g.) Muandet et al. 2014 and Chang et al. arXiv:1705.08525).  For these particular Stein effect results to be valuable the authors would need to explain---either theoretically, or just heuristically but with convincing numerical examples---why the results for an iid noise model should be of value to practitioners trying to model highly correlated (essentially, 'functional') data.
- second: the restriction to a setting in which estimation of a data-driven bandwidth is not included in the empirical Bayesian solution feels like an important limitation relative to the frequency of this additional calibration step in practice.


**Summary Of The Paper:**

In their manuscript, "Monotonicity and Double Descent in Uncertainty Estimation with Gaussian Processes", the authors consider the topic known in machine learning as 'double descent' which speaks to an important question for anyone using spectral representations of Gaussian processes for predictive modelling: how many features should I use?  A particular emphasis is given here to the performance of these models in terms of uncertainty quantification. Theoretical results are derived and numerical examples are given in some well- and miss- specified settings.


**Summary Of The Review:**

The topic is of importance and the results seem correct, but there is missing a strength of argument for the value of the insights given by these results.

---

> ### Author Response · Authors · 2022-11-07
> **Response to Reviewer ZFnV**
>
> We thank the reviewer for taking the time to read our paper and providing insightful feedback. Below, we hope to address some of your remaining concerns.
>
> **Stein effect:** The reviewer is correct that our work is related to the Stein effect — this is an older and much coarser characterisation of the double-descent phenomenon. These initial papers recognise that the inclusion of regularization reveals uniformly improved error. Double descent is more precise: implicit regularization of the model realizes itself in a way that produces a (surprising) decreasing error curve in model complexity. Double descent (and the Stein effect) has been extremely well-studied for kernel regression and random feature models (see references for Liao et al. and Liu et al. in the paper). Our aim is to extend this analysis to marginal likelihood and other predictive uncertainty metrics, which take into account the distributional properties of GPs as well. Thank you for drawing our attention to this literature, we have now included these earlier works as part of our literature review, and made the link to the Stein effect explicit.
>
> **Best-case scenario:** You are also correct that the “best-case scenario” assumption appears to be strong. For precisely this reason, we have included several real-world examples on standard benchmarks (and are happy to include more, since they are easy to produce and highly repeatable). An argument is given in the Supplementary Material to explain why our results will always carry over to practice. Essentially, appropriate whitening of any reasonable functional data compensates for model misspecification in a way that causes the mean function to decay as dimension increases.
>
> **Optimized bandwidth:** We do not consider an optimized bandwidth since the monotonicity result does not require this. Undoubtedly, optimizing this bandwidth as well would result in further improved performance. Our results do require that the bandwidth scales appropriately with dimension, which is known to occur when choosing bandwidths in practice (we refer to the discussion in El Karoui, for example).
>
> **Consequences for practitioners:** Thank you for your feedback here, we acknowledge that the practical consequences of our work deserved further clarification, and so we have updated the conclusion to outline these in further detail.
>
> **Spectral features vs input dimension**: Regarding the reviewer’s comment about the ambiguity of the term ‘dimension’: in all cases, the dimension should refer to the size of the input domain, and never to the number of the spectral features of the model (although the two may coincide, e.g. linear kernel). Please let us know if there is a use of the term that suggests otherwise and we will correct it.

---

> > ### Comment · Reviewer_ZFnV · 2022-11-09
> > **The more I think about this paper the less convinced I am of the connection to practice**
> >
> > I thank the authors for their detailed response to my review.  Indeed I had misunderstood a point on the pre-whitening of X, and unfortunately now that I do understand it I find this actually decreases my expectation that any of these results offer value in practice.  If I now understand correctly, it is not only the theoretical setting in which the X carry no information about the Y, but actually the practical experiments are designed to achieve this as well via the pre-whitening? In this case all we are seeing are some mathematical properties of the marginal likelihood of GP models applied to settings in which they offer no value; i.e., the optimal model would simply be an iid Gaussian.

---

> > > ### Author Response · Authors · 2022-11-09
> > > **Whitening is standard practice**
> > >
> > > We are a bit confused by this comment. Are you suggesting that by whitening the data, dependence of the Y on the X is removed? Whitening is standard practice in many disciplines and does not remove any functional relationship in the data, nor does it make the data Gaussian. The optimal model is most certainly not Gaussian.
> > >
> > > Random matrix theory states that the error curves can have arbitrarily many spurious “bumps” depending on the eigenspectrum of the covariance of X, but the general trend will remain the same. This is covered in Figure 14, and is precisely why the double descent literature typically treats the case of unit covariance, as this ensures there is a sensible baseline.
> > >
> > > If the reviewer is still uncomfortable with whitening, we have now run the real-data experiments without whitening the data, and reported them towards the end of Appendix A in the Supplementary Material. As expected, the curves are similar, but the behavior is less pronounced, with some extra bumps. Note that this also uses the explicit expression for lambda that we report in the theorem; these curves are likely to be smoother if lambda is optimized numerically, as is typically done in practice.
> > >
> > > We have designed our real-data experiments to align with common recommended practice. If this is not the case, please let us know, as the real-world experiments should not be synthetic in any way.

---

> > > > ### Comment · Reviewer_ZFnV · 2022-11-09
> > > > **rolling discussion**
> > > >
> > > > Thanks for the quick reply; hopefully if we have a quick back & forth it gets to the point; at the risk of it looking to the outside like we're having a twitter fight :-)
> > > >
> > > > So first of all, to make sure we're on the same page: in the conditions for Thm 1 you suppose that for each i, Y_i ~ N(0,1) such that Y_i is independent of X_i, right? Meaning that for the purpose of simulating a new Y_i there no need to know its corresponding X_i, nor the X_j and/or Y_j for any other j!=i?  That is, this is the hypothetical setting of using white noise inputs to predict white noise responses.  In a typical GP model we would rather assume that there is a functional dependence between the X_i and Y_i such that not only do we need to know X_i to simulate a new Y_i, but we'll also need to know at least some number (possibly all) the other Y_j and X_j pairs.  This is the sense in which I mean that the 'optimal' model is Gaussian: that to simulate perfectly from the generating distribution of a new Y_i we just need a source of N(0,1) variates.  The fact that we can achieve improvements by using a correlated model would in this way seem to me the connection to Stein shrinkage.
> > > >
> > > > Regarding the whitening operation on X: the proposal of ZCA (or the similar, Cholesky) whitening---a 'decorrelating' whitening---will serve to separate the datapoints more uniformly around the d-dimensional hyper-sphere than they might otherwise be in their natural form.  Certainly as compared with just standardising each d element in X independently by its sample standard deviation.  If the covariates were just white noise to begin with this is no real difference, but if they included relevant predictors that were correlated, those datapoints will now appear further away from each other in terms of a kernel for which there is a single bandwidth parameter.  (Check: this is the case right? That your bandwidth is a scalar, not a d-dimensional 'automatic relevance determination' type kernel parameter? I am familiar with pre-whitening covariates to improve causal graph discovery algorithms, and I could imagine that pre-whitening might improve automatic bandwidth selection under an appropriate shrinkage term.). In a high dimensional space everything becomes far from everything else, which is the curse of dimensionality for GPs, so I don't see why it would pay off in terms of model performance?

---

> > > > > ### Author Response · Authors · 2022-11-09
> > > > > **Continued discussion**
> > > > >
> > > > > Thank you for engaging, we appreciate the opportunity and welcome the continued discussion. In our view, a lot of the novelty of this work is precisely because it is counterintuitive, but this also invites skepticism.  To clarify: yes, the bandwidth is a scalar (scaling linearly with dimension), and our whitening procedure was ZCA.
> > > > >
> > > > > You are correct that we have assumed a well specified prior in Theorem 1 -  that $Y =  m(X) + \epsilon$ where $m$ is the prior mean, and $\epsilon$ is iid white noise. Under the marginal likelihood, this turns out to be equivalent to assuming that $m = 0$ so that $Y$ does not depend on $X$ at all. This admits fine-grained analysis on the behavior of the error curves, including an explicit asymptotic representation of the optimal regularizer.
> > > > >
> > > > > Intuitively, the ideal model should be Gaussian i.e., sending lambda to infinity in our setup. This is not reflected in the metrics we have studied. As you have identified, this is the Stein effect, but while it suggests an optimal regularizer should exist, it is not clear whether the optimal marginal likelihood still suffers the curse of dimensionality. The fact that not only does it not deteriorate, but in fact provably **improves** with dimension is the significance of Theorem 1.
> > > > >
> > > > > Our attention then largely turns to examining the behavior without these assumptions. We argue in Appendix B that unless Y explodes with dimension, there are inherent limitations on how the data-generating process behaves under whitening. This means that the marginal likelihood is a small perturbation away from the “best case” considered in Theorem 1. Our results will certainly not apply in circumstances where X and Y are not normalized (see the prior misspecification section in Appendix A), but it doesn’t seem fair to compare results across dimensions when the inputs/outputs themselves are growing or shrinking with dimension.
> > > > >
> > > > > Now the question becomes whether this perturbation is small enough to ensure that monotonicity would still hold in practice. Any assumptions on the data-generating process are unlikely to be convincing, so we appeal to experiments with real datasets for this. Note that if we do not use the optimal regularizer, we virtually never see a decreasing error curve. This curve is very sensitive to the regularizer, but **seems to occur for every dataset we try it on**. Is the decreasing error curve a strange quirk of marginal likelihood? Perhaps, but **every** metric we consider also appears to decrease when dimension becomes very large. In fact, this phenomenon seems most counterintuitive for the marginal likelihood, as it is famously used because it incorporates the curse of dimensionality.

---

### Official Review · Reviewer_Qcnt · 2022-10-24

**Confidence:** 2
**Correctness:** 3
**Technical Novelty And Significance:** 3
**Empirical Novelty And Significance:** 2
**Recommendation:** 6

**Clarity, Quality, Novelty And Reproducibility:**

I admit to not being very familiar with certain aspects of random matrix theory that are here used to derive key propositions and theorems; however, the assumptions made appeared to be sensible, as well as the overall findings and conclusions. The contributions also appear to be novel as they focus on the less explored area of how input dimensionality impacts uncertainty quantification in non-parametric models such as Gaussian processes.

The paper is well-written and the structure is sensible, although I feel as though the paper would benefit from more illustrative examples, especially in the introductory sections where concepts such as double descent may be less intuitive to grasp for readers who are unfamiliar with related work on the subject.

**Strength And Weaknesses:**

- There is always great value in papers that contribute towards demystifying how widely-used models behave under controlled set-ups, and the insights provided in this paper are expected to be of particular interest to researchers working on Gaussian processes and Bayesian deep learning models.
- I particularly appreciated how the authors considered different measurements of uncertainty quantification, and illustrated how the expectations for how learning improves with added dimensionality vary in each case. While this entails that some of the findings are less definitive, the work comes across as more complete as a result of their inclusion.
- Although the findings rely on making certain assumptions on the choice of kernel, I believe that the findings are still sufficiently broad, and it makes sense to leave further extensions, such as adapting to NTKs, for future work.
- One aspect I would have liked to see more of is how the findings of this submission influence future work on Gaussian processes and other Bayesian models? Are there any specific takeaways which might guide the design of models yielding improved estimates with better uncertainty calibration?
- Furthermore, although the authors highlight the importance of additional covariates being Gaussian for some of the derived theorems to hold, it wasn’t clear whether there are also any actionable learnings practitioners could take from this.

**Summary Of The Paper:**

There has recently been renewed interest in further studying the theoretical foundations influencing the quality of increasingly parametric models under conditions of larger dataset sizes and higher dimensionality. This paper focus on the latter case, with particular focus on how conditions observed for learning curves on predictive performance carry over to metrics that instead describe uncertainty quantification (UQ). By focusing on non-parametric Gaussian process models, the authors illustrate how increased dimensionality impacts the quality of uncertainty estimates returned by such models. With some assumptions on the choice of kernel used and the nature of additional dimensions added to existing data, the authors are able to formalize how the aforementioned UQ metrics are expected to change as input dimensionality is increased. These claims are verified using a variety of experiments on synthetic and real-world datasets.

**Summary Of The Review:**

I consider this to be a solid paper, which delivers interesting theoretic insight that could inspire further research in this direction. I do think that the paper could benefit from certain concepts being better illustrated overall however, for which some rewriting may be necessary.

---

> ### Author Response · Authors · 2022-11-07
> **Response to Reviewer Qcnt**
>
> We thank the reviewer for their careful reading, valuable feedback, and overall positive assessment of our work.
>
> **Consequences for practitioners:** Thank you for your feedback here, we acknowledge that the practical consequences of our work deserved further clarification, and so we have updated the conclusion to outline these in further detail.
>
> **Theoretical follow up work:** While the theory of learning curves, in particular double-descent curves, is now a well-understood field, to our knowledge, our result is the first to study some of the natural corresponding UQ metrics in a Bayesian perspective. As such this is a fertile area for further research to which we hope others will also contribute. We have included references to excellent review papers on double-descent in the relevant section for readers who may be unfamiliar with the phenomena. In particular, as follow up theoretical work in the GP literature, as the reviewer points out, we would like to consider kernels which do not fit under the El Karoui limit, e.g. NTK, which will require a separate analysis. We have tried to make these points clearer in the updated Conclusion.
>
> **Illustrative example in the introduction:** Thank you for the valuable suggestion. We understand that double descent curves may not be widely known yet, so an illustration early in the paper is warranted. Therefore, we have added an additional figure to the second page which illustrates the difference between a monotone curve (that we observe for Bayes free energy) and a double descent curve.

---

### Official Review · Reviewer_Qeva · 2022-10-24

**Confidence:** 3
**Clarity, Quality, Novelty And Reproducibility:** Clear and theoretically interesting w…
**Correctness:** 2
**Technical Novelty And Significance:** 2
**Empirical Novelty And Significance:** 2
**Recommendation:** 3

**Strength And Weaknesses:**

**Strengths:** The paper adequately addresses the literature around the considered question. The formulation of the problem around the marginal likelihood of GPs and predictive posterior quantities seems correct to me and the obtained theoretical result seems clear to me. Some connections are built, for instance with Fong & Holmes, 2020 and the analysis with respect to the obtained result for the regularisation hyperparameter $\lambda$ and temperature $\gamma$ is also clear.

**Weaknesses:** The manuscript presents several weaknesses that I want to remark in the following lines:

- *On GPs, Bayesian statistics and clarity:* Despite the correctness of the question addressed and the considered ideas, I think that the presentation of the GP model plus Bayesian metrics, i.e. posterior predictive and log-posterior predictive density, is far from being clear or easily understandable. This is problematic as this has been longly studied in the recent literature since Rassmussen & Williams (2006), and there are simpler and clearer ways to make the presentation of such model. Example can easily found in the recent literature. Additionally, some use of notation is not entirely positive for me, for instance, the regularisation parameter $\lambda$ appears in the whole manuscript, when it could be simply added as a kernel scale parameter, right?

- *Imprecise claims and sentences:* I find somehow some imprecise comments and claims in the paper that make me doubt or be unsure of the technical claims around the GP analysis. For instance:

> In practice, the hyperparameters (...) are often tuned to minimize the Bayes free energy. This is an
empirical Bayes procedure, and typically achieves excellent results (Krivoruchko & Gribov, 2019)

Exhaustive efforts for training GP models (including tuning of hyperparameters) based on the log-marginal likelihood have been done in the last 15 years. Sentences as the previous one, are somehow surprising to me. Even if the Bayes free energy is defined as the negative log-marginal likelihood...

- *Connections with CV, but no further analysis:* The results presented by Fong & Holmes (2020) were extremely revelant for the community and particularly for the connection between log-marginal likelihood, log-predictive scores and average k-fold-out cross validation (CV). This connection is mentioned before Section 2.3. and where the average $n^{-1}F_n$ is first described. However, to my eyes, this seems extremely related to the Theorem 1, but no mentions or links to Fong & Holmes (2020) are added. I really do not perceive much of a difference between the two results, and make me think if some this analysis is not partially covered in the previous work.

- *Analysis in a very reduced scenario:* The theoretical result is interesting to me, but seems quite reduced or constrained for GPs, when it seems could be considered from a wider perspective of more general Bayesian models. Additionally, I find the practical utility missing or at least not mentioned or properly discussed.



**Summary Of The Paper:**

The paper addresses the question that considers if the marginal likelihood and the posterior predictive losses exhibit a monotone error curve (as other learning models do) and double descent in the input dimension. For that reason, the paper introduces the necessary conditions for proving the results and verify empirically on synthetic problems that this is indeed true or not.

**Summary Of The Review:**

The paper has several flaws that make me doubt about its quality to be accepted in its current format.

---

> ### Author Response · Authors · 2022-11-07
> **Response to Reviewer Qeva**
>
> We again thank the reviewer for taking the time to read and review our paper.  There do appear to be several key misunderstandings that we would like to address. The reviewer has given us a particularly low score, however we are not entirely sure what specific objections are being raised, and what actionable improvements can be made based on the feedback. We hope this can be clarified with further discussion, which we are eager to engage in.
>
> **Summary:** The summary of our paper is incorrect. The claim that other learning models exhibit monotone error curves in the input dimension is generally false or unknown, and this is the novelty of our work. No other papers to our knowledge, and in particular none of the papers you reference, have established a monotonically increasing curve in the marginal likelihood or with respect to input dimension for any class of models. We eagerly look forward to any relevant reference that the reviewer can provide to support their assertion. In fact, it is assumed that the marginal likelihood should decrease with input dimension in most cases. The reviewer has also reported a focus on synthetic problems. This is not true. The major experiments were confirmed on the CIFAR-10 and MNIST standard real-world benchmarks. The synthetic covariates experiment was reported using UCI data.
>
> **Clarity:** There are a variety of notations for Gaussian processes, but we have mostly followed Rasmussen & Williams. However, we modified some aspects of the notation to facilitate the proof exposition. We can refer to $\lambda$ as a kernel scale parameter as this has no influence on our results, but the parameter itself cannot be ignored as it is a key quantity in our results. We remark that Reviewers Qcnt and ZFnV have complemented our exposition for being clear and well-written, but would be happy to follow any specific recommendations you have.
>
> **Imprecise claims:** Respectfully, we could not understand the essence of the reviewer’s claims as we find them rather imprecise and vague, themselves. Are there any particular issues that the reviewer can mention? The sentence used as an example is correct to our knowledge and is in line with your comments; tuning these hyperparameters has been done in this way over the last 15 years, and we refer to (Krivoruchko & Gribov, 2019) as a review. The Bayes free energy is indeed defined to be the negative-log marginal likelihood, which is how it is introduced in Section 2.2.
>
> **Connections with CV:** We agree that Fong & Holmes provides a valuable link between marginal likelihood and cross-validation. This connection means that our results imply monotonicity of averaged (over p) leave-p-out CV. However, no further connection is obvious. This work is completely different from Fong & Holmes, with distinct objectives and completely disjoint analyses. Our work shows monotonic behavior of the marginal likelihood, while Fong & Holmes connect marginal likelihood to cross-validation.
>
> **Reduced scenario:** In order to obtain these precise asymptotics, we have focused on the class of GPs, because the marginal likelihood is explicitly computable. GPs remain a valuable, useful, and broad enough class of models for practitioners and theoreticians alike. The recently discovered connections between neural networks and GPs further solidifies the importance of this class of models. Our analysis cannot be immediately extended beyond GPs, but validates current practice for parameter tuning in this setting. Obtaining similar results for more general classes of Bayesian models is a significant open problem in Bayesian deep learning, and is unlikely to hold in general.

---

### Author Response · Authors · 2022-11-07
**Comment to all reviewers**

We would like to thank all of the reviewers and the area chair for their time and consideration taken to review our work. We have responded individually to each of the reviewers and hope to clarify any issues that may have been highlighted. We have also incorporated several suggestions into a new revision of the manuscript; changes have been demarcated as blue text.

Multiple reviewers seemed to be in agreement that the practical consequences of the paper should be explained in more detail. We have now included a short list of notable consequences of this nature in the Conclusion.

---

> ### Author Response · Authors · 2022-11-18
> **Rebuttal revision**
>
> Dear reviewers,
>
> We have uploaded some final edits before the author rebuttal deadline, introducing the double descent curve in the introduction more clearly to assist readers not familiar with the term. Although we would like to provide a complete introduction to the double descent phenomenon, we are concerned about this being an effective use of the allowed space, as the field is particularly mature in the theoretical space at this point, and there is far too much to cover meaningfully. Plenty of excellent and lengthy review articles are available, which we have cited in the literature review.
>
> Heading into the next discussion phase, we remind the reviewers that we have updated the conclusion to make the practical contributions more clear. However, our work remains a theoretical contribution to the literature, and we strongly believe that theory has intrinsic value. We hope that the reviewers might have the opportunity to examine our proof to properly assess the validity of our results. There are highly cited recent works developing double descent results in kernel regression (e.g. Liang et al, 2020 and Liu et al, 2021), even at this very conference (Holzmuller, 2020). We have extended these analyses to Gaussian processes and the uncertainty metrics associated with them. The criticisms raised about practicality can be perhaps even more easily raised for much of the supporting literature.

---

### Decision · Program_Chairs · 2023-01-20

**Decision:**

Reject

**Justification For Why Not Higher Score:**

This paper explores double descent phenomena in gaussian processes. There is by now a large literature on double descent. Originally the motivation was to shed light on practical findings about why increasing the dimension could actually, counter-intuitively, improve performance. However it is not clear what the practical relevance is of the particular models and findings here.

**Justification For Why Not Lower Score:**

N/A

**Metareview: Summary, Strengths And Weaknesses:**

This paper asks: Are there double descent phenomena when performing uncertainty quantification with gaussian processes? They provide both positive and negative answers. On the negative side, they show that the learning curve is monotone with respect to the input dimension. On the positive side, they show that other forms of posterior predictive loss do exhibit double descent. The reviewers were skeptical about whether this work furnishes any insights relevant for practice.

**Summary Of Ac-Reviewer Meeting:**

N/A